# Variational Autoencoder for Deep Learning of Images, Labels and Captions

**Yunchen Pu**[†]**, Zhe Gan**[†]**, Ricardo Henao**[†]**, Xin Yuan**[‡]**, Chunyuan Li**[†]**, Andrew Stevens**[†]
**and Lawrence Carin**[†]
[†]Department of Electrical and Computer Engineering, Duke University
{yp42, zg27, r.henao, cl319, ajs104, lcarin}@duke.edu
[‡]Nokia Bell Labs, Murray Hill
xyuan@bell-labs.com

## Abstract

A novel variational autoencoder is developed to model images, as well as associated labels or captions. The Deep Generative Deconvolutional Network (DGDN) is used as a decoder of the latent image features, and a deep Convolutional Neural Network (CNN) is used as an image encoder; the CNN is used to approximate a *distribution* for the latent DGDN features/code. The latent code is also linked to generative models for labels (Bayesian support vector machine) or captions (recurrent neural network). When predicting a label/caption for a new image at test, averaging is performed across the distribution of latent codes; this is computationally efficient as a consequence of the learned CNN-based encoder. Since the framework is capable of modeling the image in the presence/absence of associated labels/captions, a new semi-supervised setting is manifested for CNN learning with images; the framework even allows *unsupervised* CNN learning, based on images alone.

## 1 Introduction

Convolutional neural networks (CNNs) [1] are effective tools for image analysis [2], with most CNNs trained in a supervised manner [2, 3, 4]. In addition to being used in image classifiers, image features learned by a CNN have been used to develop models for image captions [5, 6, 7]. Most recent work on image captioning employs a CNN for image encoding, with a recurrent neural network (RNN) employed as a decoder of the CNN features, generating a caption.

While large sets of labeled and captioned images have been assembled, in practice one typically encounters far more images without labels or captions. To leverage the vast quantity of these latter images (and to tune a model to the specific unlabeled/uncaptioned images of interest at test), semi-supervised learning of image features is of interest. To account for unlabeled/uncaptioned images, it is useful to employ a generative image model, such as the recently developed Deep Generative Deconvolutional Network (DGDN) [8, 9]. However, while the CNN is a feedforward model for image features (and is therefore fast at test time), the original DGDN implementation required relatively expensive inference of the latent image features. Specifically, in [8] parameter learning and inference are performed with Gibbs sampling or Monte Carlo Expectation-Maximization (MCEM).

We develop a new variational autoencoder (VAE) [10] setup to analyze images. The DGDN [8] is used as a decoder, and the encoder for the *distribution* of latent DGDN parameters is based on a CNN (termed a "recognition model" [10, 11]). Since a CNN is used within the recognition model, test-time speed is much faster than that achieved in [8]. The VAE framework manifests a novel means of semi-supervised CNN learning: a Bayesian SVM [12] leverages available image labels, the DGDN models the images (with or without labels), and the CNN manifests a fast encoder for the distribution of latent codes. For image-caption modeling, latent codes are shared between the CNN encoder,

DGDN decoder, and RNN caption model; the VAE learns all model parameters jointly. These models are also applicable to images alone, yielding an *unsupervised* method for CNN learning.

Our DGDN-CNN model for images is related to but distinct from prior convolutional variational auto-encoder networks [13, 14, 15]. In those models the pooling process in the encoder network is *deterministic* (max-pooling), as is the unpooling process in the decoder [14] (related to upsampling [13]). Our model uses *stochastic* unpooling, in which the unpooling map (upsampling) is inferred from the data, by maximizing a variational lower bound.

Summarizing, the contributions of this paper include: ($i$) a new VAE-based method for deep deconvolutional learning, with a CNN employed within a recognition model (encoder) for the posterior distribution of the parameters of the image generative model (decoder); ($ii$) demonstration that the fast CNN-based encoder applied to the DGDN yields accuracy comparable to that provided by Gibbs sampling and MCEM based inference, while being much faster at test time; ($iii$) the first semi-supervised CNN classification results, applied to large-scale image datasets; and ($iv$) extensive experiments on image-caption modeling, in which we demonstrate the advantages of jointly learning the image features and caption model (we also present semi-supervised experiments for image captioning).

## 2 Variational Autoencoder Image Model

### 2.1 Image Decoder: Deep Deconvolutional Generative Model

Consider $N$ images $\{\mathbf{X}^{(n)}\}_{n=1}^N$, with $\mathbf{X}^{(n)} \in \mathbb{R}^{N_x \times N_y \times N_c}$; $N_x$ and $N_y$ represent the number of pixels in each spatial dimension, and $N_c$ denotes the number of color bands in the image ($N_c = 1$ for gray-scale images and $N_c = 3$ for RGB images).

To introduce the image decoder (generative model) in its simplest form, we first consider a decoder with $L = 2$ layers. The code $\{\mathbf{S}^{(n,k_2,2)}\}_{k_2=1}^{K_2}$ feeds the decoder at the top (layer 2), and at the bottom (layer 1) the image $\mathbf{X}^{(n)}$ is generated:

$$\text{Layer 2:} \quad \tilde{\mathbf{S}}^{(n,2)} = \sum_{k_2=1}^{K_2} \mathbf{D}^{(k_2,2)} * \mathbf{S}^{(n,k_2,2)} \tag{1}$$

$$\text{Unpool:} \quad \mathbf{S}^{(n,1)} \sim \text{unpool}(\tilde{\mathbf{S}}^{(n,2)}) \tag{2}$$

$$\text{Layer 1:} \quad \tilde{\mathbf{S}}^{(n,1)} = \sum_{k_1=1}^{K_1} \mathbf{D}^{(k_1,1)} * \mathbf{S}^{(n,k_1,1)} \tag{3}$$

$$\text{Data Generation:} \quad \mathbf{X}^{(n)} \sim \mathcal{N}(\tilde{\mathbf{S}}^{(n,1)}, \alpha_0^{-1}\mathbf{I}) \tag{4}$$

Equation (4) is meant to indicate that $\mathbb{E}(\mathbf{X}^{(n)}) = \tilde{\mathbf{S}}^{(n,1)}$, and each element of $\mathbf{X}^{(n)} - \mathbb{E}(\mathbf{X}^{(n)})$ is iid zero-mean Gaussian with precision $\alpha_0$.

Concerning notation, expressions with two superscripts, $\mathbf{D}^{(k_l,l)}$, $\mathbf{S}^{(n,l)}$ and $\tilde{\mathbf{S}}^{(n,l)}$, for layer $l \in \{1, 2\}$ and image $n \in \{1, \ldots, N\}$, are 3D tensors. Expressions with three superscripts, $\mathbf{S}^{(n,k_l,l)}$, are 2D activation maps, representing the $k_l$th "slice" of 3D tensor $\mathbf{S}^{(n,l)}$; $\mathbf{S}^{(n,k_l,l)}$ is the spatially-dependent activation map for image $n$, dictionary element $k_l \in \{1, \ldots, K_l\}$, at layer $l$ of the model. Tensor $\mathbf{S}^{(n,l)}$ is formed by spatially aligning and "stacking" the $\{\mathbf{S}^{(n,k_l,l)}\}_{k_l=1}^{K_l}$. Convolution $\mathbf{D}^{(k_l,l)} * \mathbf{S}^{(n,k_l,l)}$ between 3D $\mathbf{D}^{(k_l,l)}$ and 2D $\mathbf{S}^{(n,k_l,l)}$ indicates that each of the $K_{l-1}$ 2D "slices" of $\mathbf{D}^{(k_l,l)}$ is convolved with the spatially-dependent $\mathbf{S}^{(n,k_l,l)}$; upon aligning and "stacking" these convolutions, a *tensor output* is manifested for $\mathbf{D}^{(k_l,l)} * \mathbf{S}^{(n,k_l,l)}$ (that tensor has $K_{l-1}$ 2D slices).

Assuming dictionary elements $\{\mathbf{D}^{(k_l,l)}\}$ are known, along with the precision $\alpha_0$. We now discuss the generative process of the decoder. The layer-2 activation maps $\{\mathbf{S}^{(n,k_2,2)}\}_{k_2=1}^{K_2}$ are the code that enters the decoder. Activation map $\mathbf{S}^{(n,k_2,2)}$ is spatially convolved with $\mathbf{D}^{(k_2,2)}$, yielding a 3D tensor; summing over the $K_2$ such tensors manifested at layer-2 yields the pooled 3D tensor $\tilde{\mathbf{S}}^{(n,2)}$. Stochastic *unpooling* (discussed below) is employed to go from $\tilde{\mathbf{S}}^{(n,2)}$ to $\mathbf{S}^{(n,1)}$. Slice $k_1$ of $\mathbf{S}^{(n,1)}$, $\mathbf{S}^{(n,k_1,1)}$, is convolved with $\mathbf{D}^{(k_1,1)}$, and summing over $k_1$ yields $\mathbb{E}(\mathbf{X}^{(n)})$.

For the stochastic unpooling, $\mathbf{S}^{(n,k_1,1)}$ is partitioned into contiguous $p_x \times p_y$ pooling blocks (analogous to pooling blocks in CNN-based activation maps [1]). Let $\mathbf{z}_{i,j}^{(n,k_1,1)} \in \{0,1\}^{p_x p_y}$ be a vector of $p_x p_y - 1$ zeros, and a single one; $\mathbf{z}_{i,j}^{(n,k_1,1)}$ corresponds to pooling block $(i, j)$ in $\mathbf{S}^{(n,k_1,1)}$. The

location of the non-zero element of $\boldsymbol{z}_{i,j}^{(n,k_1,1)}$ identifies the location of the *single* non-zero element in the corresponding pooling block of $\mathbf{S}^{(n,k_1,1)}$. The non-zero element in pooling block $(i,j)$ of $\mathbf{S}^{(n,k_1,1)}$ is set to $\tilde{S}_{i,j}^{(n,k_1,2)}$, *i.e.*, element $(i,j)$ in slice $k_1$ of $\tilde{\mathbf{S}}^{(n,2)}$. Within the *prior* of the decoder, we impose $\boldsymbol{z}_{i,j}^{(n,k_1,1)} \sim \mathrm{Mult}(1; 1/(p_x p_y), \ldots, 1/(p_x p_y))$. Both $\tilde{\mathbf{S}}^{(n,2)}$ and $\mathbf{S}^{(n,2)}$ are 3D tensors with $K_1$ 2D slices; as a result of the unpooling, the 2D slices in the *sparse* $\mathbf{S}^{(n,2)}$ have $p_x p_y$ times more elements than the corresponding slices in the *dense* $\tilde{\mathbf{S}}^{(n,2)}$.

The above model may be replicated to constitute $L > 2$ layers. The decoder is represented concisely as $p_{\boldsymbol{\alpha}}(\mathbf{X}|\boldsymbol{s}, \boldsymbol{z})$, where vector $\boldsymbol{s}$ denotes the "unwrapped" set of top-layer features $\{\mathbf{S}^{(\cdot,k_L,L)}\}$, and vector $\boldsymbol{z}$ denotes the unpooling maps at all $L$ layers. The model parameters $\boldsymbol{\alpha}$ are the set of dictionary elements at the $L$ layers, as well as the precision $\alpha_0$. The prior over the code is $p(\boldsymbol{s}) = \mathcal{N}(\mathbf{0}, \mathbf{I})$.

## 2.2 Image Encoder: Deep CNN

To make explicit the connection between the proposed CNN-based encoder and the above decoder, we also initially illustrate the encoder with an $L = 2$ layer model. While the two-layer decoder in (1)-(4) is top-down, starting at layer 2, the encoder is bottom-up, starting at layer 1 with image $\mathbf{X}^{(n)}$:

$$\text{Layer 1:} \quad \tilde{\mathbf{C}}^{(n,k_1,1)} = \mathbf{X}^{(n)} *_s \mathbf{F}^{(k_1,1)} \ , \ \ k_1 = 1, \ldots, K_1 \tag{5}$$

$$\text{Pool:} \quad \mathbf{C}^{(n,1)} \sim \mathrm{pool}(\tilde{\mathbf{C}}^{(n,1)}) \tag{6}$$

$$\text{Layer 2:} \quad \tilde{\mathbf{C}}^{(n,k_2,2)} = \mathbf{C}^{(n,1)} *_s \mathbf{F}^{(k_2,2)} \ , \ \ k_2 = 1, \ldots, K_2 \tag{7}$$

$$\text{Code Generation:} \quad \boldsymbol{s}_n \sim \mathcal{N}\left(\boldsymbol{\mu}_{\boldsymbol{\phi}}(\tilde{\mathbf{C}}^{(n,2)}), \mathrm{diag}(\boldsymbol{\sigma}_{\boldsymbol{\phi}}^2(\tilde{\mathbf{C}}^{(n,2)}))\right) \tag{8}$$

Image $\mathbf{X}^{(n)}$ and filter $\mathbf{F}^{(k_1,1)}$ are each tensors, composed of $N_c$ stacked 2D images ("slices"). To implement $\mathbf{X}^{(n)} *_s \mathbf{F}^{(k_1,1)}$, the respective spatial slices of $\mathbf{X}^{(n)}$ and $\mathbf{F}^{(k_1,1)}$ are convolved; the results of the $N_c$ convolutions are aligned spatially and *summed*, yielding a single 2D spatially-dependent filter output $\tilde{\mathbf{C}}^{(n,k_1,1)}$ (hence notation $*_s$, to distinguish $*$ in (1)-(4)).

The 2D maps $\{\tilde{\mathbf{C}}^{(n,k_1,1)}\}_{k_1=1}^{K_1}$ are aligned spatially and "stacked" to constitute the 3D tensor $\tilde{\mathbf{C}}^{(n,1)}$. Each contiguous $p_x \times p_y$ pooling region in $\tilde{\mathbf{C}}^{(n,1)}$ is stochastically pooled to constitute $\mathbf{C}^{(n,1)}$; the *posterior* pooling statistics in (6) are detailed below. Finally, the pooled tensor $\mathbf{C}^{(n,1)}$ is convolved with $K_2$ layer-2 filters $\{\mathbf{F}^{(k_2,2)}\}_{k_2=1}^{K_2}$, each of which yields the 2D feature map $\tilde{\mathbf{C}}^{(n,k_2,2)}$; the $K_2$ feature maps $\{\tilde{\mathbf{C}}^{(n,k_2,2)}\}_{k_2=1}^{K_2}$ are aligned and "stacked" to manifest $\tilde{\mathbf{C}}^{(n,2)}$.

Concerning the pooling in (6), let $\tilde{\mathbf{C}}_{i,j}^{(n,k_1,1)}$ reflect the $p_x p_y$ components in pooling block $(i,j)$ of $\tilde{\mathbf{C}}^{(n,k_1,1)}$. Using a multi-layered perceptron (MLP), this is mapped to the $p_x p_y$-dimensional real vector $\boldsymbol{\eta}_{i,j}^{(n,k_1,1)} = \mathrm{MLP}(\tilde{\mathbf{C}}_{i,j}^{(n,k_1,1)})$, defined as $\boldsymbol{\eta}_{i,j}^{(n,k_1,1)} = \mathbf{W}_1 \boldsymbol{h}$, with $\boldsymbol{h} = \tanh\left(\mathbf{W}_2 \mathrm{vec}(\tilde{\mathbf{C}}_{i,j}^{(n,k_1,1)})\right)$. The pooling vector is drawn $\boldsymbol{z}_{i,j}^{(n,k_1,1)} \sim \mathrm{Mult}(1; \mathrm{Softmax}(\boldsymbol{\eta}_{i,j}^{(n,k_1,1)}))$; as a recognition model, $\mathrm{Mult}(1; \mathrm{Softmax}(\boldsymbol{\eta}_{i,j}^{(n,k_1,1)}))$ is also treated as the *posterior* distribution for the DGDN *unpooling* in (2). Similarly, to constitute functions $\boldsymbol{\mu}_{\boldsymbol{\phi}}(\tilde{\mathbf{C}}^{(n,2)})$ and $\boldsymbol{\sigma}_{\boldsymbol{\phi}}^2(\tilde{\mathbf{C}}^{(n,2)})$ in (8), each layer of $\tilde{\mathbf{C}}^{(n,2)}$ is fed through a distinct MLP. Details are provided in the Supplementary Material (SM).

Parameters $\boldsymbol{\phi}$ of $q_{\boldsymbol{\phi}}(\boldsymbol{s}, \boldsymbol{z}|\mathbf{X})$ correspond to the filter banks $\{\mathbf{F}^{(k_l,l)}\}$, as well as the parameters of the MLPs. The encoder is a CNN (yielding fast testing), utilized in a novel manner to manifest a *posterior* distribution on the parameters of the decoder. As discussed in Section 4, the CNN is trained in a novel manner, allowing *semi-supervised* and even *unsupervised* CNN learning.

## 3 Leveraging Labels and Captions

### 3.1 Generative Model for Labels: Bayesian SVM

Assume a label $\ell_n \in \{1, \ldots, C\}$ is associated with training image $\mathbf{X}^{(n)}$; in the discussion that follows, labels are assumed available for each image (for notational simplicity), but in practice only a subset of the $N$ training images need have labels. We design $C$ one-versus-all binary SVM classifiers

[16], responsible for mapping top-layer image features $\boldsymbol{s}_n$ to label $\ell_n$; $\boldsymbol{s}_n$ is the same image code as in (8), from the top DGDN layer. For the $\ell$-th classifier, with $\ell \in \{1, \ldots, C\}$, the problem may be posed as training with $\{\boldsymbol{s}_n, y_n^{(\ell)}\}_{n=1}^N$, with $y_n^{(\ell)} \in \{-1, 1\}$. If $\ell_n = \ell$ then $y_n^{(\ell)} = 1$, and $y_n^{(\ell)} = -1$ otherwise. Henceforth we consider the Bayesian SVM for each one of the binary learning tasks, with labeled data $\{\boldsymbol{s}_n, y_n\}_{n=1}^N$.

Given a feature vector $\boldsymbol{s}$, the goal of the SVM is to find an $f(\boldsymbol{s})$ that minimizes the objective function $\gamma \sum_{n=1}^N \max(1 - y_n f(\boldsymbol{s}_n), 0) + R(f(\boldsymbol{s}))$, where $\max(1 - y_n f(\boldsymbol{s}_n), 0)$ is the hinge loss, $R(f(\boldsymbol{s}))$ is a regularization term that controls the complexity of $f(\boldsymbol{s})$, and $\gamma$ is a tuning parameter controlling the trade-off between error penalization and the complexity of the classification function. Recently, [12] showed that for the linear classifier $f(\boldsymbol{s}) = \boldsymbol{\beta}^T \boldsymbol{s}$, minimizing the SVM objective function is equivalent to estimating the mode of the pseudo-posterior of $\boldsymbol{\beta}$: $p(\boldsymbol{\beta}|\mathbf{S}, \boldsymbol{y}, \gamma) \propto \prod_{n=1}^N \mathcal{L}(y_n|\boldsymbol{s}_n, \boldsymbol{\beta}, \gamma)p(\boldsymbol{\beta}|\cdot)$, where $\boldsymbol{y} = [y_1 \ \ldots \ y_N]^T$, $\mathbf{S} = [\boldsymbol{s}_1 \ \ldots \ \boldsymbol{s}_N]$, $\mathcal{L}(y_n|\boldsymbol{s}_n, \boldsymbol{\beta}, \gamma)$ is the pseudo-likelihood function, and $p(\boldsymbol{\beta}|\cdot)$ is the prior distribution for the vector of coefficients $\boldsymbol{\beta}$. In [12] it was shown that $\mathcal{L}(y_n|\boldsymbol{s}_n, \boldsymbol{\beta}, \gamma)$ admits a location-scale mixture of normals representation by introducing latent variables $\lambda_n$:

$$\mathcal{L}(y_n|\boldsymbol{s}_n, \boldsymbol{\beta}, \gamma) = e^{-2\gamma \max(1 - y_n \boldsymbol{\beta}^T \boldsymbol{s}_n, 0)} = \int_0^\infty \frac{\sqrt{\gamma}}{\sqrt{2\pi\lambda_n}} \exp\left(-\frac{(1 + \lambda_n - y_n \boldsymbol{\beta}^T \boldsymbol{s}_n)^2}{2\gamma^{-1}\lambda_n}\right) d\lambda_n. \qquad (9)$$

Note that (9) is a mixture of Gaussian distributions w.r.t. random variable $y_n \boldsymbol{\beta}^T \boldsymbol{s}_n$, where the mixture is formed with respect to $\lambda_n$, which controls the mean and variance of the Gaussians. This encourages data augmentation for variable $\lambda_n$, permitting efficient Bayesian inference (see [12, 17] for details).

Parameters $\{\boldsymbol{\beta}_\ell\}_{\ell=1}^C$ for the $C$ binary SVM classifiers are analogous to the fully connected parameters of a softmax classifier connected to the top of a traditional CNN [2]. If desired, the pseudo-likelihood of the SVM-based classifier can be replaced by a softmax-based likelihood. In Section 5 we compare performance of the SVM and softmax based classifiers.

## 3.2 Generative Model for Captions

For image $n$, assume access to an associated caption $\mathbf{Y}^{(n)}$; for notational simplicity, we again assume a caption is available for each training image, although in practice captions may only be available on a subset of images. The caption is represented as $\mathbf{Y}^{(n)} = (\boldsymbol{y}_1^{(n)}, \ldots, \boldsymbol{y}_{T_n}^{(n)})$, and $\boldsymbol{y}_t^{(n)}$ is a 1-of-$V$ ("one-hot") encoding, with $V$ the size of the vocabulary, and $T_n$ the length of the caption for image $n$. Word $t$, $\boldsymbol{y}_t^{(n)}$, is embedded into an $M$-dimensional vector $\boldsymbol{w}_t^{(n)} = \mathbf{W}_e \boldsymbol{y}_t^{(n)}$, where $\mathbf{W}_e \in \mathbb{R}^{M \times V}$ is a word embedding matrix (to be learned), *i.e.*, $\boldsymbol{w}_t^{(n)}$ is a column of $\mathbf{W}_e$, chosen by the one-hot $\boldsymbol{y}_t^{(n)}$.

The probability of caption $\mathbf{Y}^{(n)}$ given top-layer DGDN image features $\boldsymbol{s}_n$ is defined as $p(\mathbf{Y}^{(n)}|\boldsymbol{s}_n) = p(\boldsymbol{y}_1^{(n)}|\boldsymbol{s}_n) \prod_{t=2}^{T_n} p(\boldsymbol{y}_t^{(n)}|\boldsymbol{y}_{<t}^{(n)}, \boldsymbol{s}_n)$. Specifically, we generate the first word $\boldsymbol{y}_1^{(n)}$ from $\boldsymbol{s}_n$, with $p(\boldsymbol{y}_1^{(n)}) = \mathrm{softmax}(\mathbf{V}\boldsymbol{h}_1^{(n)})$, where $\boldsymbol{h}_1^{(n)} = \tanh(\mathbf{C}\boldsymbol{s}_n)$. Bias terms are omitted for simplicity. All other words in the caption are then sequentially generated using a recurrent neural network (RNN), until the end-sentence symbol is generated. Each conditional $p(\boldsymbol{y}_t^{(n)}|\boldsymbol{y}_{<t}^{(n)}, \boldsymbol{s}_n)$ is specified as $\mathrm{softmax}(\mathbf{V}\boldsymbol{h}_t^{(n)})$, where $\boldsymbol{h}_t^{(n)}$ is recursively updated through $\boldsymbol{h}_t^{(n)} = \mathcal{H}(\boldsymbol{w}_{t-1}^{(n)}, \boldsymbol{h}_{t-1}^{(n)})$. $\mathbf{C}$ and $\mathbf{V}$ are weight matrices (to be learned), and $\mathbf{V}$ is used for computing a distribution over words.

The transition function $\mathcal{H}(\cdot)$ can be implemented with a *gated* activation function, such as Long Short-Term Memory (LSTM) [18] or a Gated Recurrent Unit (GRU) [19]. Both LSTM and GRU have been proposed to address the issue of learning long-term dependencies. In experiments we have found that GRU provides slightly better performance than LSTM (we implemented and tested both), and therefore the GRU is used.

## 4 Variational Learning of Model Parameters

To make the following discussion concrete, we describe learning and inference within the context of images and captions, combining the models in Sections 2 and 3.2. This learning setup is also applied to model images with associated labels, with the caption model replaced in that case with the Bayesian SVM of Section 3.1 (details provided in the SM). In the subsequent discussion we employ the image encoder $q_{\boldsymbol{\phi}}(\boldsymbol{s}, \boldsymbol{z}|\mathbf{X})$, the image decoder $p_{\boldsymbol{\alpha}}(\mathbf{X}|\boldsymbol{s}, \boldsymbol{z})$, and the generative model for the caption (denoted $p_{\boldsymbol{\psi}}(\mathbf{Y}|\boldsymbol{s})$, where $\boldsymbol{\psi}$ represents the GRU parameters).

The desired parameters $\{\phi, \alpha, \psi\}$ are optimized by minimizing the variational lower bound. For a single captioned image, the variational lower bound $\mathcal{L}_{\phi,\alpha,\psi}(\mathbf{X}, \mathbf{Y})$ can be expressed as

$$\mathcal{L}_{\phi,\alpha,\psi}(\mathbf{X}, \mathbf{Y}) = \xi\{\mathbb{E}_{q_\phi(s|\mathbf{X})}[\log p_\psi(\mathbf{Y}|s)]\} + \mathbb{E}_{q_\phi(s,z|\mathbf{X})}[\log p_\alpha(\mathbf{X}, s, z) - \log q_\phi(s, z|\mathbf{X})]$$

where $\xi$ is a tuning parameter that balances the two components of $\mathcal{L}_{\phi,\alpha,\psi}(\mathbf{X}, \mathbf{Y})$. When $\xi$ is set to zero, it corresponds to the variational lower bound for a single uncaptioned image:

$$\mathcal{U}_{\phi,\alpha}(\mathbf{X}) = \mathbb{E}_{q_\phi(s,z|\mathbf{X})}[\log p_\alpha(\mathbf{X}, s, z) - \log q_\phi(s, z|\mathbf{X})] \tag{10}$$

The lower bound for the entire dataset is then:

$$\mathcal{J}_{\phi,\alpha,\psi} = \sum_{(\mathbf{X},\mathbf{Y}) \in \mathcal{D}_c} \mathcal{L}_{\phi,\alpha,\psi}(\mathbf{X}, \mathbf{Y}) + \sum_{\mathbf{X} \in \mathcal{D}_u} \mathcal{U}_{\phi,\alpha}(\mathbf{X}) \tag{11}$$

where $\mathcal{D}_c$ denotes the set of training images with associated captions, and $\mathcal{D}_u$ is the set of training images that are uncaptioned (and unlabeled).

To optimize $\mathcal{J}_{\phi,\alpha,\psi}$ w.r.t. $\phi$, $\psi$ and $\alpha$, we utilize Monte Carlo integration to approximate the expectation, $\mathbb{E}_{q_\phi(s,z|\mathbf{X})}$, and stochastic gradient descent (SGD) for parameter optimization. We use the variance reduction techniques in [10] and [11] to compute the gradients. Details are provided in the SM.

When $\xi$ is set to 1, $\mathcal{L}_{\phi,\alpha,\psi}(\mathbf{X}, \mathbf{Y})$ recovers the exact variational lower bound. Motivated by assigning the same weight to every data point, we set $\xi = N_X/(T\rho)$ or $N_X/(C\rho)$ in the experiments, where $N_X$ is the number of pixels in each image, $T$ is the number of words in the corresponding caption, $C$ is the number of categories for the corresponding label and $\rho$ is the proportion of labeled/captioned data in the mini-batch.

At test time, we consider two tasks: inference of a caption or label for a new image $\mathbf{X}^\star$. Again, considering captioning of a new image (with similar inference for labeling), after the model parameters are learned $p(\mathbf{Y}^\star|\mathbf{X}^\star) = \int p_\psi(\mathbf{Y}^\star|s^\star)p(s^\star|\mathbf{X}^\star)ds^\star \approx \sum_{s=1}^{N_s} p_\psi(\mathbf{Y}^\star|s_s^\star)$, where $s_s^\star \sim q_\phi(s|\mathbf{X} = \mathbf{X}^\star)$, and $N_s$ is the number of samples. Monte Carlo sampling is used to approximate the integral, and the recognition model, $q_\phi(s|\mathbf{X})$, is employed to approximate $p(s|\mathbf{X})$, for fast inference of image representation.

## 5 Experiments

The architecture of models and initialization of model parameters are provided in the SM. No dataset-specific tuning other than early stopping on validation sets was conducted. The Adam algorithm [20] with learning rate 0.0002 is utilized for optimization of the variational learning expressions in Section 4. We use mini-batches of size 64. Gradients are clipped if the norm of the parameter vector exceeds 5, as suggested in [21]. All the experiments of our models are implemented in Theano [22] using a NVIDIA GeForce GTX TITAN X GPU with 12GB memory.

### 5.1 Benchmark Classification

We first present image classification results on MNIST, CIFAR-10 & -100 [23], Caltech 101 [24] & 256 [25], and ImageNet 2012 datasets. For Caltech 101 and Caltech 256, we use 30 and 60 images per class for training, respectively. The predictions are based on averaging the decision values of $N_s = 50$ collected samples from the approximate posterior distribution over the latent variables from $q_\phi(s|\mathbf{X})$. As a reference for computational cost, our model takes about 5 days to train on ImageNet.

We compared our VAE setup to a VAE with deterministic unpooling, and we also compare with a DGDN trained using Gibbs sampling and MCEM [8]; classification results and testing time are summarized in Table 1. Other state-of-the-art results can be found in [8]. The results based on Gibbs sampling and MCEM are obtained by our own implementation on the same GPU, which are consistent with the classification accuracies reported in [8].

For Gibbs-sampling-based learning, only suitable for the first five small/modest size datasets we consider, we collect 50 posterior samples of model parameters $\alpha$, after 1000 burn-in iterations during training. Given a sample of model parameters, the inference of top-layer features at test is also done via Gibbs sampling. Specifically, we collect 100 samples after discarding 300 burn-in samples; fewer samples leads to worse performance. The predictions are based on averaging the decision values

Table 1: Classification error (%) and testing time (ms per image) on benchmarks.

| **Method** | MNIST | | CIFAR-10 | | CIFAR-100 | | Caltech 101 | | Caltech 256 | |
| --- | --- | --- | --- | --- | --- | --- | --- | --- | --- | --- |
| | test error | test time | test error | test time | test error | test time | test error | test time | test error | test time |
| Gibbs [8] | 0.37 | 3.1 | 8.21 | 10.4 | 34.33 | 10.4 | 12.87 | 50.4 | 29.50 | 52.3 |
| MCEM [8] | 0.45 | 0.8 | 9.04 | 1.1 | 35.92 | 1.1 | 13.51 | 8.8 | 30.13 | 8.9 |
| VAE-d | 0.42 | **0.007** | 10.74 | **0.02** | 37.96 | **0.02** | 14.79 | **0.3** | 32.18 | **0.3** |
| VAE (Ours) | 0.38 | **0.007** | 8.19 | **0.02** | 35.01 | **0.02** | 11.99 | **0.3** | 29.33 | **0.3** |

| **Method** | ImageNet 2012 | | | ImageNet Pretrained for | | | |
| --- | --- | --- | --- | --- | --- | --- | --- |
| | | | | Caltech 101 | | Caltech 256 | |
| | top-1 error | top-5 error | test time | test error | test time | test error | test time |
| MCEM [8] | 37.9 | 16.1 | 14.4 | 6.85 | 14.1 | 22.10 | 14.2 |
| VAE (Ours) | 38.2 | 15.7 | **1.0** | 6.91 | **0.9** | 22.53 | **0.9** |

of the collected samples (50 samples of model parameters $\alpha$, and for each 100 inference samples of latent parameters $s$ and $z$, for a total of 5000 samples). With respect to the testing of MCEM, all data-dependent latent variables are integrated (summed) out in the expectation, except for the top-layer feature map, for which we find a MAP point estimate via gradient descent.

As summarized in Table 1, the proposed recognition model is *much* faster than Gibbs sampling and MCEM at test time (up to 400x speedup), and yields accuracy commensurate with these other two methods (often better). To illustrate the role of stochastic unpooling, we replaced it with deterministic unpooling as in [14]. The results, indicated as VAE-d in Table 1, demonstrate the powerful capabilities of the stochastic unpooling operation. We also tried VAE-d on the ImageNet 2012 dataset; however, the performance is much worse than our proposed VAE, hence those results are not reported.

## 5.2 Semi-Supervised Classification

We now consider semi-supervised classification. With each mini-batch, we use 32 labeled samples and 32 unlabeled samples, *i.e.*, $\rho = 0.5$.

Table 2: Semi-supervised classification error (%) on MNIST. $N$ is the number of labeled images per class.

| $N$ | TSVM | Deep generative model [26] | | Ladder network [27] | | Our model | |
| --- | --- | --- | --- | --- | --- | --- | --- |
| | | M1+TSVM | M1+M2 | $\Gamma$-full | $\Gamma$-conv | $\xi = 0$ | $\xi = N_x/(C\rho)$ |
| 10 | 16.81 | 11.82± 0.25 | 3.33 ± 0.14 | 1.06 ± 0.37 | **0.89**±0.50 | 5.83 ± 0.97 | 1.49 ± 0.36 |
| 60 | 6.16 | 5.72± 0.05 | 2.59 ±0.05 | - | 0.82 ± 0.17* | 2.19 ± 0.19 | **0.77** ± 0.09 |
| 100 | 5.38 | 4.24± 0.07 | 2.40 ±0.02 | 0.84 ± 0.08 | 0.74 ± 0.10* | 1.75 ± 0.14 | **0.63** ± 0.06 |
| 300 | 3.45 | 3.49± 0.04 | 2.18 ±0.04 | - | 0.63 ± 0.02* | 1.42 ± 0.08 | **0.51** ± 0.04 |

*These results are achieved with our own implementation based on the publicly available code.

**MNIST**   We first test our model on the MNIST classification benchmark. We randomly split the 60,000 training samples into a 50,000-sample training set and a 10,000-sample validation set (used to evaluate early stopping). The training set is further randomly split into a labeled and unlabeled set, and the number of labeled samples in each category varies from 10 to 300. We perform testing on the standard 10,000 test samples with 20 different training-set splits.

Table 2 shows the classification results. For $\xi = 0$, the model is trained in an *unsupervised* manner. When doing unsupervised learning, the features extracted by our model are sent to a separate transductive SVM (TSVM). In this case, our results can be directly compared to the results of the M1+TSVM model [26], demonstrating the effectiveness of our recognition model in providing good representations of images. Using 10 labeled images per class, our semi-supervised learning approach with $\xi = N_x/(C\rho)$ achieves a test error of 1.49, which is competitive with state-of-the-art results [27]. When using a larger number of labeled images, our model consistently achieves the best results.

**ImageNet 2012**   ImageNet 2012 is used to assess the scalability of our model to large datasets (also considered, for supervised learning, in Table 1). Since no comparative results exist for semi-supervised learning with ImageNet, we implemented the 8-layer AlexNet [2] and the 22-layer GoogLeNet [4] as the supervised model baselines, which were trained by utilizing only the labeled data[1]. We split the 1.3M training images into a labeled and unlabeled set, and vary the proportion

of labeled images from 1% to 100%. The classes are balanced to ensure that no particular class is over-represented, *i.e.*, the ratio of labeled and unlabeled images is the same for each class. We repeat the training process 10 times, and each time we utilize different sets of images as the unlabeled ones.

Figure 1 shows our results, together with the baselines. Tabulated results and a plot with error bars are provided in the SM. The variance of our model's results (caused by different randomly selected labeled examples) is around 1% when considering a small proportion of labeled images (less than 10% labels), and the variance drops to less than 0.2% when the proportion of labeled images is larger than 30%. As can be seen from Figure 1, our semi-supervised learning approach with 60% labeled data achieves comparable results (61.24% top-1 accuracy) with the results of full datasets (61.8% top-1 accuracy), demonstrating the effectiveness of our approach for semi-supervised classification. Our model provides consistently better results than AlexNet [2] which has a similar five convolutional layers architecture as ours. Our model is outperformed by GoogLeNet when more labeled images are provided. This is not surprising since GoogLeNet utilizes a considerably more complicated CNN architecture than ours.

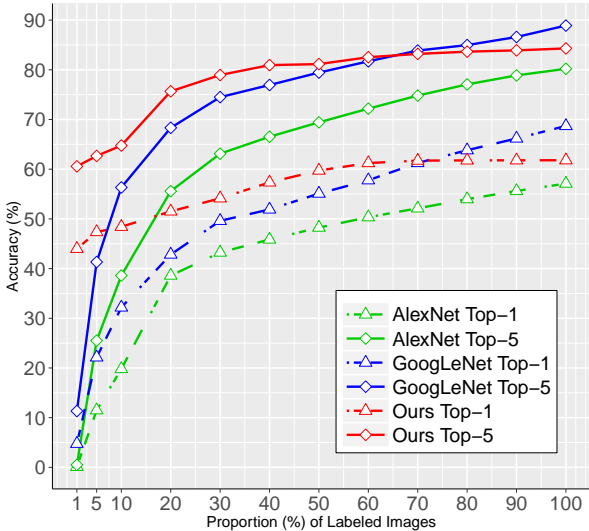

Figure 1: Semi-supervised classification accuracy on the validation set of ImageNet 2012.

To further illustrate the role of each component of our model, we replaced the Bayesian SVM with a softmax classifier (see discussion at the end of Section 3.1). The softmax results are slightly worse, and provided in the SM. The gap between the results of Bayesian SVM and softmax are around 1% when the proportion of labeled images is less 30% and drop to around 0.5% when a larger proportion of labeled images is considered (larger than 30%). This further illustrates that the performance gain is primarily due to the semi-supervised learning framework used in our model, rather than the discriminative power of the SVM.

## 5.3   Image Captioning

We present image captioning results on three benchmark datasets: Flickr8k [29], Flickr30k [30] and Microsoft (MS) COCO [31]. These datasets contain 8000, 31000 and 123287 images, respectively. Each image is annotated with 5 sentences. For fair comparison, we use the same pre-defined splits for all the datasets as in [5]. We use 1000 images for validation, 1000 for test and the rest for training on Flickr8k and Flickr30k. For MS COCO, 5000 images are used for both validation and testing. The widely used BLEU metric [32] and sentence perplexity ($\mathcal{PPL}$) are employed to quantitatively evaluate the performance of our image captioning model. A low $\mathcal{PPL}$ indicates a better language model. For the MS COCO dataset, we further evaluate our model with metrics METEOR [33] and CIDEr [34]. Our joint model takes three days to train on MS COCO.

We show results for three models: ($i$) *Two-step model*: this model consists of our generative and recognition model developed in Section 2 to analyze images alone, in an *unsupervised* manner. The extracted image features are fed to a separately trained RNN. ($ii$) *Joint model*: this is the joint model developed in Sections 2 and 3.2. ($iii$) *Joint model with ImageNet*: in this model training is performed in a semi-supervised manner, with the training set of ImageNet 2012 treated as uncaptioned images, to complement the captioned training set.

The image captioning results are summarized in Table 3. Our two-step model achieves better performance than similar baseline two-step methods, in which VggNet [3] and GoogLeNet [4] were used as feature extractors. The baseline VggNet and GoogLeNet models require labeled images for training, and hence are trained on ImageNet. By contrast, in our two-step approach, the deep model is trained in an *unsupervised* manner, using uncaptioned versions of images from the training set. This fact may explain the improved quality of our results in Table 3.

Table 3: BLEU-1,2,3,4, METEOR, CIDEr and $\mathcal{PPL}$ metrics compared to other state-of-the-art results and baselines on Flickr8k, Flickr 30k and MS COCO datasets.

| Method | Flickr8k | | | | | Flickr30k | | | | |
|---|---|---|---|---|---|---|---|---|---|---|
| | B-1 | B-2 | B-3 | B-4 | $\mathcal{PPL}$ | B-1 | B-2 | B-3 | B-4 | $\mathcal{PPL}$ |
| *Baseline results* | | | | | | | | | | |
| VggNet+RNN | 0.56 | 0.37 | 0.24 | 0.16 | 15.71 | 0.57 | 0.38 | 0.25 | 0.17 | 18.83 |
| GoogLeNet+RNN | 0.56 | 0.38 | 0.24 | 0.16 | 15.71 | 0.58 | 0.39 | 0.26 | 0.17 | 18.77 |
| Our two step model | 0.61 | 0.41 | 0.27 | 0.17 | 15.82 | 0.61 | 0.41 | 0.27 | 0.17 | 18.73 |
| *Our results with other state-of-the-art results* | | | | | | | | | | |
| Hard-Attention [6] | 0.67 | 0.46 | 0.31 | 0.21 | - | 0.67 | 0.44 | 0.30 | 0.20 | - |
| Our joint model | 0.70 | 0.49 | 0.33 | 0.22 | 15.24 | 0.69 | 0.50 | 0.35 | 0.22 | 16.17 |
| Our joint model with ImageNet | 0.72 | 0.52 | 0.36 | 0.25 | 13.24 | 0.72 | 0.53 | 0.38 | 0.25 | 15.34 |
| *State-of-the-art results using extra information* | | | | | | | | | | |
| Attributes-CNN+RNN [7] | 0.74 | 0.54 | 0.38 | 0.27 | 12.60 | 0.73 | 0.55 | 0.40 | 0.28 | 15.96 |

| Method | MS COCO | | | | | | |
|---|---|---|---|---|---|---|---|
| | B-1 | B-2 | B-3 | B-4 | METEOR | CIDEr | $\mathcal{PPL}$ |
| *Baseline results* | | | | | | | |
| VggNet+RNN | 0.61 | 0.42 | 0.28 | 0.19 | 0.19 | 0.56 | 13.16 |
| GoogLeNet+RNN | 0.60 | 0.40 | 0.26 | 0.17 | 0.19 | 0.55 | 14.01 |
| Our two step | 0.61 | 0.42 | 0.27 | 0.18 | 0.20 | 0.58 | 13.46 |
| *Our results with other state-of-the-art results* | | | | | | | |
| DMSM [28] | - | - | - | 0.26 | 0.24 | - | 18.10 |
| Hard-Attention [6] | 0.72 | 0.50 | 0.36 | 0.25 | 0.23 | - | - |
| Our joint model | 0.71 | 0.51 | 0.38 | 0.26 | 0.22 | 0.89 | 11.57 |
| Our joint model with ImageNet | 0.72 | 0.52 | 0.37 | 0.28 | 0.24 | 0.90 | 11.14 |
| *State-of-the-art results using extra information* | | | | | | | |
| Attributes-CNN+LSTM [7] | 0.74 | 0.56 | 0.42 | 0.31 | 0.26 | 0.94 | 10.49 |

It is worth noting that our *joint* model yields significant improvements over our two-step model, nearly 10% in average for BLEU scores, demonstrating the importance of inferring a shared latent structure. It can also be seen that our improvement with semi-supervised use of ImageNet is most significant with the small/modest datasets (Flickr8k and Flickr30k), compared to the large dataset (MS COCO). Our model performs better than most image captioning systems. The only method with better performance than ours is [7], which employs an intermediate image-to-attributes layer, that requires determining an extra attribute vocabulary. Examples of generated captions from the validation set of ImageNet 2012, *which has no ground truth captions and is unseen during training* (the semi-supervised learning only uses the training set of ImageNet 2012), are shown in Figure 2.

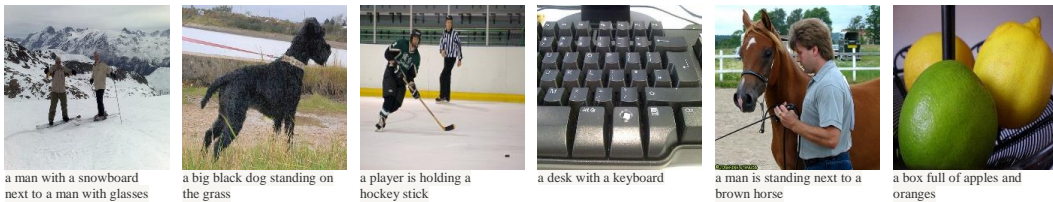

a man with a snowboard next to a man with glasses
a big black dog standing on the grass
a player is holding a hockey stick
a desk with a keyboard
a man is standing next to a brown horse
a box full of apples and oranges

Figure 2: Examples of generated caption from unseen images on the validation dataset of ImageNet.

# 6 Conclusions

A recognition model has been developed for the Deep Generative Deconvolutional Network (DGDN) [8], based on a novel use of a deep CNN. The recognition model has been coupled with a Bayesian SVM and an RNN, to also model associated labels and captions, respectively. The model is learned using a variational autoencoder setup, and allows semi-supervised learning (leveraging images without labels or captions). The algorithm has been scaled up with a GPU-based implementation, achieving results competitive with state-of-the-art methods on several tasks (and novel semi-supervised results).

# Acknowledgements

This research was supported in part by ARO, DARPA, DOE, NGA, ONR and NSF. The Titan X used in this work was donated by the NVIDIA Corporation.

## Footnotes

[1]We use the default settings in the Caffe package, which provide a top-1 accuracy of 57.1% and 68.7%, as well as a top-5 accuracy of 80.2% and 88.9% on the validation set for AlexNet and GoogLeNet, respectively.

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
