[Supplementary Material · NIPS supplementary material.pdf]

# Variational Autoencoder for Deep Learning of Images, Labels and Captions: Supplementary Material

**Yunchen Pu[†], Zhe Gan[†], Ricardo Henao[†], Xin Yuan[‡], Chunyuan Li[†], Andrew Stevens[†] and Lawrence Carin[†]**
[†]Department of Electrical and Computer Engineering, Duke University
{yp42, zg27, r.henao, cl319, ajs104, lcarin}@duke.edu
[‡]Nokia Bell Labs, Murray Hill
xyuan@bell-labs.com

## 1 Semi-supervised Results on ImageNet 2012

Table 1: Semi-supervised classification accuracy (%) on the validation set of ImageNet 2012.

| Proportion | 1% | 5% | 10% | 20% | 30% | 40% |
|---|---|---|---|---|---|---|
| *top-1* | | | | | | |
| AlexNet | $0.1 \pm 0.01$ | $11.5 \pm 0.72$ | $19.8 \pm 0.71$ | $38.6 \pm 0.31$ | $43.23 \pm 0.28$ | $45.85 \pm 0.23$ |
| GoogeLeNet | $4.75 \pm 0.58$ | $22.13 \pm 1.14$ | $32.18 \pm 0.80$ | $42.83 \pm 0.28$ | $49.61 \pm 0.11$ | $51.90 \pm 0.20$ |
| BSVM (ours) | $43.98 \pm 1.15$ | $47.36 \pm 0.91$ | $48.41 \pm 0.76$ | $51.51 \pm 0.28$ | $54.14 \pm 0.12$ | $57.34 \pm 0.18$ |
| Softmax (ours) | 42.89 | 46.42 | 47.51 | 50.75 | 53.49 | 56.83 |
| *top-5* | | | | | | |
| AlexNet | $0.5 \pm 0.01$ | $25.5 \pm 0.92$ | $38.60 \pm 0.90$ | $55.58 \pm 0.25$ | $63.12 \pm 0.23$ | $66.53 \pm 0.22$ |
| GoogeLeNet | $11.33 \pm 0.96$ | $41.33 \pm 1.34$ | $56.33 \pm 0.86$ | $68.33 \pm 0.21$ | $74.50 \pm 0.12$ | $76.94 \pm 0.14$ |
| Ours | $60.57 \pm 1.61$ | $62.67 \pm 1.14$ | $64.76 \pm 0.90$ | $75.67 \pm 0.19$ | $78.95 \pm 0.10$ | $80.94 \pm 0.13$ |
| Softmax (ours) | 59.20 | 61.40 | 63.58 | 74.96 | 78.39 | 80.46 |
| | | | | | | |
| Proportion | 50% | 60% | 70% | 80% | 90% | 100% |
| *top-1* | | | | | | |
| AlexNet | $48.25 \pm 0.23$ | $50.34 \pm 0.18$ | $52.12 \pm 0.14$ | $53.97 \pm 0.14$ | $55.62 \pm 0.09$ | 57.1 |
| GoogeLeNet | $55.09 \pm 0.23$ | $57.78 \pm 0.23$ | $61.25 \pm 0.15$ | $63.82 \pm 0.17$ | $66.18 \pm 0.05$ | 68.7 |
| BSVM (ours) | $59.73 \pm 0.21$ | $61.24 \pm 0.19$ | $61.72 \pm 0.14$ | $61.77 \pm 0.13$ | $61.79 \pm 0.04$ | 61.8 |
| Softmax (ours) | 59.33 | 60.91 | 61.40 | 61.44 | 61.49 | 61.53 |
| *top-5* | | | | | | |
| AlexNet | $69.43 \pm 0.18$ | $72.18 \pm 0.19$ | $74.81 \pm 0.13$ | $77.06 \pm 0.13$ | $78.87 \pm 0.09$ | 80.2 |
| GoogeLeNet | $79.44 \pm 0.17$ | $81.70 \pm 0.11$ | $83.87 \pm 0.14$ | $84.97 \pm 0.18$ | $86.6 \pm 0.09$ | 88.9 |
| BSVM (ours) | $81.15 \pm 0.13$ | $82.53 \pm 0.10$ | $83.2 \pm 0.12$ | $83.65 \pm 0.17$ | $83.91 \pm 0.08$ | 84.3 |
| Softmax (ours) | 80.68 | 82.12 | 82.82 | 83.13 | 83.51 | 83.88 |

Figure 1: Semi-supervised classification accuracy on the validation set of ImageNet 2012.

Table 2: Architecture of the image models. Image Size: spatial size $\times$ color channel (one for gray and three for RGB), *e.g.*, $28^2 \times 1$. Dictionary: dictionary number $\times$ dictionary spatial size, *e.g.*, $30 \times 8^2$. Pooling: pooling/unpooling window size, *e.g.*, $3 \times 3$.

| Dataset | Image Size | | Layer-1 | Layer-2 | Layer-3 | Layer-4 | Layer-5 |
|---|---|---|---|---|---|---|---|
| | | | \multicolumn Model Architecture | | | | |
| MNIST | $28^2 \times 1$ | Dictionary | $30 \times 8^2$ | $80 \times 6^2$ | - | - | - |
| | | Pooling | $3 \times 3$ | - | - | - | - |
| CIFAR-10 | $32^2 \times 3$ | Dictionary | $48 \times 5^2$ | $128 \times 5^2$ | $128 \times 5^2$ | - | - |
| | | Pooling | $2 \times 2$ | $2 \times 2$ | - | - | - |
| CIFAR-100 | $32^2 \times 3$ | Dictionary | $48 \times 5^2$ | $128 \times 5^2$ | $128 \times 5^2$ | - | - |
| | | Pooling | $2 \times 2$ | $2 \times 2$ | - | - | - |
| Caltech 101 | $128^2 \times 3$ | Dictionary | $48 \times 7^2$ | $84 \times 5^2$ | $84 \times 5^2$ | - | - |
| | | Pooling | $4 \times 4$ | $2 \times 2$ | - | - | - |
| Caltech 256 | $128^2 \times 3$ | Dictionary | $48 \times 7^2$ | $128 \times 5^2$ | $128 \times 5^2$ | - | - |
| | | Pooling | $4 \times 4$ | $2 \times 2$ | - | - | - |
| ImageNet | $256^2 \times 3$ | Dictionary | $96 \times 5^2$ | $256 \times 5^2$ | $512 \times 5^2$ | $1024 \times 5^2$ | $512 \times 5^2$ |
| | | Pooling | $4 \times 4$ | $2 \times 2$ | $2 \times 2$ | $2 \times 2$ | - |
| Flickr8k | $256^2 \times 3$ | Dictionary | $48 \times 5^2$ | $84 \times 5^2$ | $128 \times 5^2$ | $192 \times 5^2$ | $128 \times 5^2$ |
| | | Pooling | $4 \times 4$ | $2 \times 2$ | $2 \times 2$ | $2 \times 2$ | - |
| Flickr30k | $256^2 \times 3$ | Dictionary | $48 \times 5^2$ | $84 \times 5^2$ | $128 \times 5^2$ | $384 \times 5^2$ | $256 \times 5^2$ |
| | | Pooling | $4 \times 4$ | $2 \times 2$ | $2 \times 2$ | $2 \times 2$ | - |
| MS COCO | $256^2 \times 3$ | Dictionary | $48 \times 5^2$ | $84 \times 5^2$ | $128 \times 5^2$ | $512 \times 5^2$ | $384 \times 5^2$ |
| | | Pooling | $4 \times 4$ | $2 \times 2$ | $2 \times 2$ | $2 \times 2$ | - |

## 2 Model Architecture and Initialization

The architecture of the image models for each dataset in all the experiments are summarized in Table 2. For example, MNIST data is composed of gray images with spatial size $28 \times 28$ and CIFAR-10 is composed of RGB color images with spatial size $32 \times 32$. A two-layer model is used with dictionary element size $8 \times 8$ and $6 \times 6$ at the first and second layer, respectively. The pooling size is $3 \times 3$ ($p_x = p_y = 3$) and the number of dictionary elements at layers 1 and 2 are $K_1 = 30$ and $K_2 = 80$, respectively.

All the parameters for the image model are initialized at random and we do not perform layer-wise pretraining as in [1]. For the RNN training employed in image captioning, we initialize all recurrent matrices with orthogonal initialization as suggested in [2]. Non-recurrent weights are initialized from an uniform distribution in [-0.01,0.01]. All the bias terms are initialized to zero. Word vectors are initialized with the publicly available *word2vec* vectors that were trained on 100 billion words from Google News, these vectors have dimensionality 300 and were trained using a continuous bag-of-words architecture [3]. Words not present in the set of pretrained words are initialized at random. The number of hidden units in the RNNs is set to 512.

## 3 Details for the Variational Autoencoder

### 3.1 Image Captioning

Recall the variational lower bound for image captioning:

$$\mathcal{L}(\mathbf{X}, \mathbf{Y}) = \xi\big\{\mathbb{E}_{q_\phi(\boldsymbol{s}|\mathbf{X})}[\log p_\psi(\mathbf{Y}|\boldsymbol{s})]\big\} + \mathbb{E}_{q_\phi(\boldsymbol{s},\boldsymbol{z}|\mathbf{X})}[\log p_\alpha(\mathbf{X},\boldsymbol{s},\boldsymbol{z}) - \log q_\phi(\boldsymbol{s},\boldsymbol{z}|\mathbf{X})] \quad (1)$$

The gradient of the variational lower bound w.r.t to the decoder model parameters is straightforward:

$$\nabla_\psi \mathcal{L}(\mathbf{X}, \mathbf{Y}) = \xi\mathbb{E}_{q_\phi(\boldsymbol{s}|\mathbf{X})}[\nabla_\psi \log p_\psi(\mathbf{Y}|\boldsymbol{s})] \quad (2)$$

$$\nabla_\alpha \mathcal{L}(\mathbf{X}, \mathbf{Y}) = \mathbb{E}_{q_\phi(\boldsymbol{s},\boldsymbol{z}|\mathbf{X})}[\nabla_\alpha \log p_\alpha(\mathbf{X}|\boldsymbol{s},\boldsymbol{z})] \quad (3)$$

The corresponding gradient w.r.t the encoder model is

$$\nabla_\phi \mathcal{L}(\mathbf{X}, \mathbf{Y}) = \xi\big\{\mathbb{E}_{q_\phi(\boldsymbol{s}|\mathbf{X})}[\log p_\psi(\mathbf{Y}|\boldsymbol{s})] \times \nabla_\phi \log q_\phi(\boldsymbol{s}|\mathbf{X})\big\}$$
$$+ \mathbb{E}_{q_\phi(\boldsymbol{s},\boldsymbol{z}|\mathbf{X})}\big\{[\log p_\alpha(\mathbf{X}|\boldsymbol{s},\boldsymbol{z}) - \log q_\phi(\boldsymbol{s},\boldsymbol{z}|\mathbf{X})] \times \nabla_\phi \log q_\phi(\boldsymbol{s},\boldsymbol{z}|\mathbf{X})\big\} \quad (4)$$

If we use Monte Carlo integration to approximate the expectation in (4), the variance of the estimator can be very high. Since there are both real and binary latent variables in (1), we use the variance reduction techniques in [4] and [5]. The variational lower bound in (1) can be expressed as

$$\mathcal{L}(\mathbf{X}, \mathbf{Y}) = \tag{5}$$
$$= \xi\{\mathbb{E}_{q_\phi(s|\mathbf{X})}[\log p_\psi(\mathbf{Y}|s)]\} + \mathbb{E}_{q_\phi(s,z|\mathbf{X})}[\log p_\alpha(\mathbf{X}, z|s) + \log p_\alpha(s) - \log q_\phi(z|\mathbf{X}) - \log q_\phi(s|\mathbf{X})]$$
$$= \xi\{\mathbb{E}_{q_\phi(s|\mathbf{X})}[\log p_\psi(\mathbf{Y}|s)]\} - D_{KL}[q_\phi(s|\mathbf{X})||p_\alpha(s)] + \mathbb{E}_{q_\phi(s,z|\mathbf{X})}[\log p_\alpha(\mathbf{X}, z|s) - \log q_\phi(z|\mathbf{X})]$$
$$= -D_{KL}[q_\phi(s|\mathbf{X})||p_\alpha(s)] + \mathbb{E}_{q_\phi(s|\mathbf{X})}\Big\{\xi[\log p_\psi(\mathbf{Y}|s)] + \mathbb{E}_{q_\phi(z|\mathbf{X})}[\log p_\alpha(\mathbf{X}, z|s) - \log q_\phi(z|\mathbf{X})]\Big\}$$

Recall that $q_\phi(s|\mathbf{X}) = \mathcal{N}(\boldsymbol{\mu}_\phi(\tilde{\mathbf{C}}^{(L)}), \text{diag}(\boldsymbol{\sigma}_\phi^2(\tilde{\mathbf{C}}^{(L)})))$ and $p(s) = \mathcal{N}(\mathbf{0}, \mathbf{I})$. Assume $J$ is the dimension of $z$, and $\mu_j$ and $\sigma_j$ is the $j$th element of $\boldsymbol{\mu}_\phi(\tilde{\mathbf{C}}^{(L)})$ and $\boldsymbol{\sigma}_\phi(\tilde{\mathbf{C}}^{(L)})$, respectively. We can get the closed form of the KL term:

$$-D_{KL}[q_\phi(s|\mathbf{X})||p_\alpha(s)] = \frac{1}{2}\sum_{j=1}^{J}\left\{(1 - (\mu_j)^2 - (\sigma_j)^2 + \log((\sigma_j)^2)\right\} \tag{6}$$

Using the reparameterization trick in [4]

$$s = f(\phi, \epsilon) = \boldsymbol{\mu}_\phi(\tilde{\mathbf{C}}^{(L)}) + \epsilon(\boldsymbol{\sigma}_\phi(\tilde{\mathbf{C}}^{(L)})), \quad \epsilon \sim \mathcal{N}(\mathbf{0}, \mathbf{I}) \tag{7}$$

The expectation term can be expressed as

$$\mathbb{E}_{q_\phi(s|\mathbf{X})}\Big\{\xi[\log p_\psi(\mathbf{Y}|s)] + \mathbb{E}_{q_\phi(z|\mathbf{X})}[\log p_\alpha(\mathbf{X}, z|s) - \log q_\phi(z|\mathbf{X})]\Big\} \tag{8}$$
$$= \mathbb{E}_{p(\epsilon)}\Big\{\xi[\log p_\psi(\mathbf{Y}|s = f(\phi, \epsilon))] + \mathbb{E}_{q_\phi(z|\mathbf{X})}[\log p_\alpha(\mathbf{X}, z|s = f(\phi, \epsilon)) - \log q_\phi(z|\mathbf{X})]\Big\}$$

Therefore, the gradient of the lower bound with respect to $\phi$ can be expressed as

$$\nabla_\phi\mathcal{L}(\mathbf{X}, \mathbf{Y}) = -\nabla_\phi D_{KL}[q_\phi(s|\mathbf{X})||p_\alpha(s)] \tag{9}$$
$$+ \mathbb{E}_{p(\epsilon)}\Big\{\nabla_\phi\xi[\log p_\psi(\mathbf{Y}|s = f(\phi, \epsilon))] \tag{10}$$
$$+ \nabla_\phi\mathbb{E}_{q_\phi(z|\mathbf{X})}[\log p_\alpha(\mathbf{X}, z|s = f(\phi, \epsilon)) - \log q_\phi(z|\mathbf{X})]\Big\} \tag{11}$$

This expectation can approximated by Monto Carlo sampling:

$$\frac{1}{N_s}\sum_{i=1}^{N_s}\Big\{\nabla_\phi\xi[\log p_\psi(\mathbf{Y}|s = f(\phi, \epsilon_i))] \tag{12}$$
$$+ \nabla_\phi\mathbb{E}_{q_\phi(z|\mathbf{X})}[\log p_\alpha(\mathbf{X}, z|s = f(\phi, \epsilon_i)) - \log q_\phi(z|\mathbf{X})]\Big\} \tag{13}$$

where $\nabla_\phi\mathbb{E}_{q_\phi(z|\mathbf{X})}[\log p_\alpha(\mathbf{X}, z) - \log q_\phi(z|\mathbf{X})]$ is the same gradient as in [5].

## 3.2 Image Classification

Recall that the pseudo-likelihood of a label $\ell_n \in \{1, \ldots, C\}$

$$\mathcal{L}(\ell_n|s_n, \boldsymbol{\beta}, \gamma) = \prod_{\ell=1}^{C}(y_n^{(\ell)}|s_n, \boldsymbol{\beta}_\ell, \gamma_\ell) \tag{14}$$

$$= \prod_{\ell=1}^{C}\left\{\int_0^\infty \frac{\sqrt{\gamma_\ell}}{\sqrt{2\pi\lambda_n^{(\ell)}}}\exp\left(-\frac{(1 + \lambda_n^{(\ell)} - y_n^{(\ell)}\boldsymbol{\beta}_\ell^T s_n)^2}{2\gamma_\ell^{-1}\lambda_n^{(\ell)}}\right)d\lambda_n^{(\ell)}\right\} \tag{15}$$

$\boldsymbol{\beta}$ is treated as another model parameter (part of $\psi$). $\lambda_n^{(\ell)}$ is treated as latent variable. We have

$$p(\ell_n, \boldsymbol{\lambda}_n|s_n, \boldsymbol{\beta}, \gamma) = \prod_{\ell=1}^{C}(y_n^{(\ell)}|s_n, \lambda_n^{(\ell)}, \boldsymbol{\beta}_\ell, \gamma_\ell) \tag{16}$$

$$= \prod_{\ell=1}^{C}\left\{\frac{\sqrt{\gamma_\ell}}{\sqrt{2\pi\lambda_n^{(\ell)}}}\exp\left(-\frac{(1 + \lambda_n^{(\ell)} - y_n^{(\ell)}\boldsymbol{\beta}_\ell^T s_n)^2}{2\gamma_\ell^{-1}\lambda_n^{(\ell)}}\right)\right\} \tag{17}$$

Therefore, the variational lower bound for image classification is

$$\mathcal{L}(\mathbf{X}, \mathbf{Y}) = \xi \{ \mathbb{E}_{q_\phi(s_n, \lambda_n | \mathbf{X}_n, \ell_n)} [\log p_\psi(\lambda_n, \ell_n | s)] \}$$
$$+ \; \mathbb{E}_{q_\phi(s, z | \mathbf{X})} [\log p_\alpha(\mathbf{X}, s, z) - \log q_\phi(s, z | \mathbf{X})] \tag{18}$$

Since for the most part (18) is the same as for the image caption model, we only discuss the gradient of the lower bound w.r.t. $\boldsymbol{\beta}$. The first term of variational lower bound can be expressed as

$$\mathbb{E}_{q_\phi(s_n, \lambda_n | \mathbf{X}_n, \ell_n)} [\log p_\psi(\lambda_n, \ell_n | s)] = \sum_{\ell=1}^{C} \mathbb{E}_{q_\phi(s_n, \lambda_n^{(\ell)} | \mathbf{X}_n, y_n^{(\ell)})} [\log p_\psi(\lambda_n^{(\ell)}, y_n^{(\ell)} | s_n)] \tag{19}$$

Note that $q_\phi(s_n, \lambda_n | \mathbf{X}_n, y_n^{(\ell)}) = q_\phi(s_n | \mathbf{X}_n) q_\phi(\lambda_n | y_n^{(\ell)})$, hence we get

$$\sum_{\ell=1}^{C} \mathbb{E}_{q_\phi(s_n, \lambda_n^{(\ell)} | \mathbf{X}_n, y_n^{(\ell)})} [\log p_\psi(\lambda_n^{(\ell)}, y_n^{(\ell)} | s_n)] \tag{20}$$

$$= \sum_{\ell=1}^{C} \mathbb{E}_{q_\phi(s_n | \mathbf{X}_n)} \left\{ \mathbb{E}_{q_\phi(\lambda_n^{(\ell)} | y_n^{(\ell)})} [\log p_\psi(\lambda_n^{(\ell)}, y_n^{(\ell)} | s_n)] \right\} \tag{21}$$

Since

$$\log p_\psi(\lambda_n^{(\ell)}, y_n^{(\ell)} | s_n) = -\frac{(1 + \lambda_n^{(\ell)} - y_n^{(\ell)} \boldsymbol{\beta}_\ell^T s_n)^2}{2 \gamma_\ell^{-1} \lambda_n^{(\ell)}} + c(\lambda_n^{(\ell)}, y_n^{(\ell)}, \gamma_\ell) \tag{22}$$

where $c(\lambda_n^{(\ell)}, y_n^{(\ell)}, \gamma_\ell)$ are independent of $\beta_\ell$, we can find that the relevant portion of Equation (22) is a linear function of $(\lambda_n^{(\ell)})^{-1}$. It means the expectation term $\mathbb{E}_{q_\phi(\lambda_n^{(\ell)} | y_n^{(\ell)})} [\log p_\psi(\lambda_n^{(\ell)}, y_n^{(\ell)} | s_n)]$ in Equation (21) can be obtained by simply replacing $(\lambda_n^{(\ell)})^{-1}$ with its conditional expectation. From [6], we have

$$q_\phi((\lambda_n^{(\ell)})^{-1} | y_n^{(\ell)}) = \mathcal{IG}(|1 - y_n^\ell s_n^\top \boldsymbol{\beta}^{(\ell)}|^{-1}, 1) \tag{23}$$
$$\mathbb{E}((\lambda_n^{(\ell)})^{-1}) = |1 - y_n^\ell s_n^\top \boldsymbol{\beta}^{(\ell)}|^{-1} \tag{24}$$

Thus, using the same reparameterization trick in (7), we can get the gradient w.r.t. $\boldsymbol{\beta}$.

## 4 Multilayer Perceptrons

$\boldsymbol{\mu}_\phi(\tilde{\mathbf{C}}^{(n,2)})$ and $\boldsymbol{\sigma}_\phi(\tilde{\mathbf{C}}^{(n,2)})$ are constituted by "stacking" the $K_2$ spatially aligned $\boldsymbol{\mu}_\phi(\tilde{\mathbf{C}}^{(n,k_2,2)})$ and $\boldsymbol{\sigma}_\phi(\tilde{\mathbf{C}}^{(n,k_2,2)})$, respectively, which are defined as (bias are omitted in the main paper)

$$\boldsymbol{\mu}_\phi(\tilde{\mathbf{C}}^{(n,k_2,2)}) = \mathbf{W}_\mu^{(k_2)} h^{(k_2)} + b_\mu^{(k_2)} \tag{25}$$
$$\log \boldsymbol{\sigma}_\phi(\tilde{\mathbf{C}}^{(n,k_2,2)}) = \mathbf{W}_\phi^{(k_2)} h^{(k_2)} + b_\phi^{(k_2)} \tag{26}$$
$$h^{(k_2)} = \tanh\left( \mathbf{W}^{(k_2)} \mathrm{vec}(\tilde{\mathbf{C}}^{(n,k_2,2)}) + b^{(k_2)} \right) \tag{27}$$

where $k_2 = 1, \ldots, K_2$.