[Reviews · NeurIPS 2016]

Reviewer 1

Summary

This paper presents a new variational autoencoder (VAE) for images, which also is capable of predicting labels and captions. The proposed framework is based on using Deep Generative Deconvolutional Networks (DGDNs) as a decoders of the latent image features, and a deep Convolutional Neural Network (CNN) as the encoder which approximates the distribution encoded by the VAE. The approach leads to a fast CNN-based encoder and experience show it can yield accuracy comparable to that provided by Gibbs sampling and MCEM based inference, while being much faster at test time. The paper claims to provide the "first semi-supervised CNN classification results, applied to large-scale image datasets" and certainly provides extensive experiments on image-caption modelling, showing benefits of the approach and exploring semi-supervised learning schemes.

Qualitative Assessment

This paper presents a model and method which are likely to be of interest to many in the community. The formulation allows stochastic layers of a convolution-deconvolution structured autoencoder with stochastic layers to be parameterized and manipulated in such a way that inference can be performed in a much more computationally efficient way compared to Gibbs sampling and MCEM techniques. Results on CIFAR are fairly far from state of the art, but illustrate the key contributions related to efficient inference well. Results for a few other methods would be useful to include in Table 1 to give the reader some context as to what regime this approach and the examined model are operating within. The Flickr8k, 30k and MS COCO results seem quite strong, especially given that this work does not focus on the application, but rather the proposed VAE method. This work does use an underlying formulation that seems quite similar to the the deep conditional variational autoencoders (CVAEs) proposed in: Yan, X., Yang, J., Sohn, K., & Lee, H. (2015). Attribute2Image: Conditional Image Generation from Visual Attributes. arXiv preprint arXiv:1512.00570. As such the work above should probably be mentioned and cited. Other work has also examined some of the general ideas here, but it is sometimes cast as simply a 'convolutional autoencoder' approach as opposed to a 'Generative Deconvolutional Network'. Work such as that of Makhzani, A., & Frey, B. J. (2015). Winner-take-all autoencoders. In Advances in Neural Information Processing Systems (pp. 2791-2799). representing one such example. As such I think some of the claims to novelty should be toned down a little. I'm thinking in particular of the line ""first semi-supervised CNN classification results, applied to large-scale image datasets", which seems like a bit of a stretch; however, I am open to the authors providing further points to back such an assertion in their response. It seems this assertion is based mainly on the fact that ImageNet results have been presented, whereas other work has not? The ImageNet results are certainly a welcome addition to this work; however, it would be even more compelling if a semi-supervised approach could be used to truly push the state of the art higher. That is of course a tall order given the scale of the labels for the complete imageNet evaluation. A few more minor points: The title would parse better if it were: "Variational Autoencoder[s] for Deep Learning [with] Images, Labels and Caption[s]" Line 5: "The latent code is also linked to generative models for labels (Bayesian support vector machine)" SVMs are not generally considered to be generative models. I'd suggest a different turn of phrase be used when describing this component of the model. "Since the framework is capable of modeling the image in the presence/absence of associated labels/captions, a new semi-supervised setting is manifested for CNN learning with image[s]". Please note that other work has proposed semi-supervised learning using auxiliary tasks, ex. Dosovitskiy, A., Springenberg, J. T., Riedmiller, M., & Brox, T. (2014). Discriminative unsupervised feature learning with convolutional neural networks. In Advances in Neural Information Processing Systems (pp. 766-774).

Confidence in this Review

2-Confident (read it all; understood it all reasonably well)


Reviewer 2

Summary

The authors propose a convolutional extension of the variational autoencoder with specific design choices for the pooling and unpooling operations. In addition, methods to jointly model image labels (using a Bayesian SVM) and captions (using a RNN) are introduced. Extensive experiments are performed on label prediction and image captioning, showing in particular the benefit of joint modeling of image/label/caption, and the ability of the model to effectively use unlabelled data.

Qualitative Assessment

This was a well written paper with multiple significant contributions and very thorough evaluation. The deconvolution VAE is pretty complicated but the exposition is for the most part very well written. The only parts I remain slightly unclear on in the pooling step are a) how is C^(n,1) generated from eta b) how is the z in the pooling and unpooling tied? Can you comment on the linear nature of the decoder? Did you find nonlinearities here were not helpful? The empirical results very clearly demonstrate the strengths of the model. I hope the authors will make code available as this contribution should be of valuable to may practicioners. Re figure 2: state whether these were randomly picked or picked by hand: they seem remarkedly good!

Confidence in this Review

2-Confident (read it all; understood it all reasonably well)


Reviewer 3

Summary

The paper in review contrasts most of previous work on deep generative models used likelihood to evaluate the performance of the models. It thus provides important practical value of using deep generative models to improve classification accuracy. Applications of deep generative models directly to improve prediction accuracy in ImageNet and MS COCO tasks are impressive. The experiments are interesting.

Qualitative Assessment

Most of previous work on deep generative models used likelihood to evaluate the performance of the models. This misses important practical value of using deep generative models to improve classification accuracy. The paper in review breaks this trend, and applies deep generative models directly to improve prediction accuracy in ImageNet and MS COCO tasks. Experiments are interesting. However, for the MS COCO task on image captioning, an important prior work which gave excellent captioning accuracy was missing as the baseline: From captions to visual concepts and back Hao Fang, Saurabh Gupta, Forrest Iandola, Rupesh K Srivastava, Li Deng, Piotr Dollár, Jianfeng Gao, Xiaodong He, Margaret Mitchell, John C Platt, C Lawrence Zitnick, Geoffrey Zweig, IEEE CVPR, 2015 The revision of the paper should include a discussion about this.

Confidence in this Review

2-Confident (read it all; understood it all reasonably well)


Reviewer 4

Summary

This paper apply the variational auto-encoder structure into image classification, semi-supervised and image caption domains. The authors did a lot work on the experiments on several datasets to prove the the validity and efficiency of their proposed method. However, there are some issues in the experiment part, which may seriously influence the final conclusion.

Qualitative Assessment

Variational autoencoder has been hotly discussed in CV domains e.g. image classification and image generation. However, the method proposed in this paper does not provide a new perspective for these domains. Although the authors did a lot work on experiments, it's incomplete. The evidences are weak and may lead to a incorrect conclusion. My personal suggestions are: 1. Complete the experiment results in MINST dataset. 2. Apply semi-supervised methods e.g. M1+M2 and Ladder networks on ImageNet2012 dataset (the same as on MNIST dataset), even if they are time-consuming.

Confidence in this Review

3-Expert (read the paper in detail, know the area, quite certain of my opinion)


Reviewer 5

Summary

This paper describes an approach to perform semi-supervised learning on images. The authors specifically consider classification and captioning tasks, although the method is applicable to other tasks that take single images as inputs. The proposed approach consists of a type of variational autoencoder (VAE) where an encoder maps images to a latent feature space and a decoder then reconstructs images based on these latent features. The latent features are then used as inputs for a classification or text generation model appropriate for the task. This architecture enables semi-supervised training, either by pre-training the autoencoder in an unsupervised way and then training the task-specific model (two-step training) or by training end-to-end the autoencoder and the task-specific model as a single model on labeled examples and only the autoencoder on unlabeled examples (joint training). For the autoencoder, the proposed approach uses a convolutional network (CNN) as the encoder, a deep generative deconvolutional network (DGDN) as the decoder, for the task-specific models it uses a Bayesian SVM for classification or a gated recurrent unit (GRU) for captioning. The paper differs from similar approaches described in the literature for its use of stochastic pooling and unpooling in the encoder and the decoder, respectively. Also, it doesn't make use of non-linear activation functions after each layer, as it is usual, rather it uses tanh activation functions to compute stochastic pooling probabilities and in the final Gaussian sampling layer in the encoder. The use of a Bayesian SVM for classification is also unusual, although, together with stochastic unpooling, it was already found in the original DGDN paper.

Qualitative Assessment

The paper is clearly written and the approach is sound, although it appears as a straightforward combination of existing approaches, specifically VAE, DGDN and image captioning by RNN. The approach includes some architectural details, specifically the stochastic pooling/unpooling and the lack of non-linear activations between convolutions in the encoder and the decoder, which differ from standard approaches described in the literature. The authors don't motivate these choices. It would be advisable if these choices were motivated by theoretical arguments or individual experimental comparisons. On the other hand, the choice of the Bayesian SVM as a classifier, while unusual, is motivated by experimental comparison against the more common softmax classifier. The experimental section evaluates the model on supervised classification, semi-supervised classification and captioning in comparison with existing methids. For supervised classification, the paper compares the proposed approach to the DGDN approaches, but not to plain supervised approaches or other semi-supervised approaches that could have provided reasonable baselines. The authors report comparable test errors and faster test speeds to the DGDN methods on standard benchmark datasets. No significance estimates are provided. Since the test errors are close, significance tests could have helped to better assess these results. For semi-supervised classification, the paper compares the proposed approach to transductive SVM, standard non-convolutional VAE-based semi-supervised classifiers and Ladder network. The MNIST dataset is used. They outperform TSVM and VAE and slightly underperform Ladder network (though still within the significance range, which is provided this time). This result is interesting, although a direct comparison with convolutional VAE should have been performed to assess the usefulness of the architectural variant introduced by this approach. They also compare their method against standard image classification CNNs on Imagenet. As a side remark,it shall be noted that neither MNIST nor Imagenet are true semi-supervised classification datasets, as in more natural scenarios, such as domain adaptation, the distributions of the labeled and unlabeled examples are often different. Experiments on native semi-supervised datasets would allow a better evaluation of the method robustness to distribution difference. For semi-supervised captioning, the paper reports experiments on standard dataset with the addition of source images from Imagenet as unlabeled examples. They compare against purely supervised approaches in the literature, obtaining good, though not SOTA results. No significance estimates are given and the table is difficult to read due to lack of bolding. In conclusion the paper presents an interesting, though not particularly novel, approach that may be valuable and does not present any obvious flaws, but more theoretical motivation or experimental validation would be advisable to assess its worth.

Confidence in this Review

2-Confident (read it all; understood it all reasonably well)


Reviewer 6

Summary

This paper couples convolutional VAEs for generative image modeling with labeling/captioning of images for semi-supervised learning. The models presented use a stochastic pooling method optimized using a variational lower bound. A series of fully and semi-supervised classification results are presented as well as a comparison of Bayesian-SVM and softmax classification methods. Lastly the authors provide results for image captioning for both a jointly and 2-stage trained model.

Qualitative Assessment

This paper provides a number of experiments for convolutional VAEs using stochastic pooling layers. Interestingly the authors use a Bayesian SVM for classification instead of the more widely used (DGM models) softmax function. With respect to novelty the Convolution-VAE model is in it self not a novel contribution since this have been used in several other papers previously, which is also pointed out by the authors. Secondly semi-supervised classification using Deep Generative Models (DGMs) have been explored previously in both Kingma et. al 2015 and Maaløe et. al. 2016. On the other hand stochastic pooling is a new and possibly interesting idea and probabilistic language models have to the best of my knowledge not been applied to captioning before. With respect to the stochastic pooling 1) How do the performance of the used convolutional-VAE compared with either a) stochastic, b) deterministic or c)noisy/random pooling? 2) I would have liked an analysis of the data dependency of the stochastic pooling layer using i.e. KL[q(s|X)|p(s)]. If this term is zero the stochastic pooling effectively defaults to pooling layer with locations randomly sampled form p(s) I believe? For the probabilistic image captioning some interesting results are presented even though the results are not as good as the current best methods. Encouragingly the jointly trained model works better than the two-step trained model and some improvements are seen by semi-supervised trained together with Imagenet. However I would like to see a more throughout analysis of the results including: 1) What are the KL[q|p]-divergences for the different model parts (Image, caption, and pooling) ? 2) Examples of sampled image / caption pairs from the generative model 3) Comparison of generated images and lower bounds for a pure generative image model as well as for the image/caption model. For the semi-supervised classification results on MNIST the Conv-VAE performs somewhat poorer than i.e. the Ladder Network using 100 labeled data. Using more labels the authors claim to have better performance. My concern here is that a) the authors do not compare with the best LadderNetwork performance (The reported 3.06% and 1.53% performance should have been 1.06% and 0.84%, table1 Rasmus 2015 et. al. I believe) and b) the LadderNetwork is permutation invariant whereas the conv-VAE utilizes spatial information? The semi-supervised Imagenet results looks good, especially for the low data regime. What are the results for similar previously published models if any? Overall the paper presents an impressive number of experiements. However I find that the analysis could be more throughout and miss important parts w.r.t analysing the probabilistic model components. Of less importance I find the paper difficult to read (especially section 2 and 3 that in my opinion contain to many details cluttering the overall message) with the text needing some more work.

Confidence in this Review

2-Confident (read it all; understood it all reasonably well)